# Open Innovation as the Catalyst in the Personalized Medicine to Personalized Digital Medicine Transition

**DOI:** 10.3390/jpm12091500

**Published:** 2022-09-14

**Authors:** Alfredo Cesario, Marika D’Oria, Irene Simone, Stefano Patarnello, Vincenzo Valentini, Giovanni Scambia

**Affiliations:** 1Scientific Directorate, Fondazione Policlinico Universitario A. Gemelli IRCCS, 00182 Rome, Italy; 2Gemelli Digital Medicine & Health, 00182 Rome, Italy; 3Gemelli Generator, Gemelli Science and Technology Park (G-STeP), Fondazione Policlinico Universitario A. Gemelli IRCCS, 00182 Rome, Italy; 4Department of Diagnostic Imaging, Radiation Oncology and Hematology, Fondazione Policlinico Universitario A. Gemelli IRCCS, 00182 Rome, Italy

**Keywords:** personalized medicine, digital medicine, open innovation, knowledge, interdisciplinarity

## Abstract

Personalized medicine (PM) bridges several disciplines for understanding and addressing prevalent, complex, or rare situations in human health (e.g., complex phenotyping, risk stratification, etc.); therefore, digital and technological solutions have been integrated in the field to boost innovation and new knowledge generation. The open innovation (OI) paradigm proposes a method by which to respectfully manage disruptive change in biomedical organizations, as experienced by many organizations during digital transformation and the COVID-19 pandemic. In this article, we focus on how this paradigm has catalyzed the transition from PM to personalized digital medicine in a large-volume research hospital. Methods, challenges, and results are discussed. This case study is an endeavor to confirm that OI strategies could help manage urgent needs from the healthcare environment, while achieving sustainability-oriented, accountable innovation.

## 1. Introduction

Personalized medicine (PM) bridges several disciplines to generate new knowledge for understanding and addressing prevalent, complex, or rare situations in human health (e.g., complex phenotyping, risk stratification, etc.) [1]. Although heterogeneous and multiple data have been generated through the years (from genomics to epigenetics), novel technologies have been developed to collect, analyze, and manage information from clinical and real-world scenarios, including artificial intelligence (AI)-based mathematical formalisms for data analysis or digital solutions for data capture [2].

The integration of digital technologies and the Internet of medical things (IoMT) within the ecosystem of data available for patient care represented an important filter for managing the heterogeneous but fundamental health parameters, which otherwise could not be retrievable except with a physical visit [3]. The necessity of a real-time remote patient monitoring (RPM) (e.g., symptoms, adherence to therapy) also led to a progressive digitalization of clinical pathways.

Despite this, organizations like large-volume hospitals still face several challenges:-Care continuity: the delocalization of healthcare services encounters new strategies of RPM (e.g., virtual assistance or technological support) that need to be integrated with the traditional one.-Business continuity: the valorization of off-the-shelf products (e.g., clinical expertise, scientific knowledge) should be conducted in a sustainable and ethical way (in terms of costs, protection of intellectual property and privacy, etc.).-Innovation continuity: the deployment of effective clinical and digital solutions must adapt to rapid changes and new healthcare challenges.

In this scenario, digital medicine (DM) arises as a discipline that aims at generating strong evidence to support the use of trustworthy technologies within the biomedical field to ameliorate therapeutic pathways, to strengthen clinical practices, to anticipate disease progression or recurrence, and to model and predict clinical outcomes from synthetic data to digital specimen collection [4]. A non-conclusive list of examples of DM products and/or services are shown in Table 1.

By assessing any possible (and potential) scientific and technological quality and effectiveness, DM necessarily involves a multi-dimensional, multi-setting dialogue among physicians, researchers, patients, caregivers, clinical care providers, regulators, policy makers, medical devices companies, data scientists, engineers, funders, etc.

As an immediate consequence of the width of the cross-fertilization involved, DM has the potential to innovate the prevention, prediction, and participation of issues pertaining to the patient during the whole care process. This “innovation boost” certainly promotes personalization too, thanks to the scientific acceleration in discovering molecular targeted therapies or the development of high-throughput biotechnologies, for the validation of biomarkers predictive of response, or the generation of accurate prognostic tools (e.g., oncology) [11].

However, the sole combination of digital solutions with -omic and clinical information does not automatically bring to a personalized DM (PDM), because the researcher/clinician should be able to understand how to use these solutions by acquiring new skills (and a whole new grammar/syntaxis) that come mostly from other disciplines and ontological domains (e.g., information technology, engineering, data analysis, etc.). 

In this article, we introduce and describe the paradigm of open innovation as a catalyst of the transition from PM to PDM within biomedical organizations. We will discuss the case study of an Italian research hospital as a measurable example of how an organization can boost its scientific and “market” impact by innovating practices, processes, products, and professionals’ skills. In addition, we will also give additional background regarding DM for the purpose of setting the desired context of reasoning.

## 2. Framework: From Biology to Open Innovation

### 2.1. How Knowledge Is Generated

Innovation can come from several cognitive inputs: the need for adaptation to a challenging situation, a gap to be fulfilled, a discovery by serendipity, an illumination by insight, dreams, chimeric ambitions, etc. All these seeds of innovation can sprout into valuable resources for research and development (R&D) if they find an environment (an organization) that takes care of them from the idea generation to the go-to-market strategy. Every organization is a creator and a carrier of knowledge, understood to be a complex phenomenon according to organizational theories because:-It can be explicit, tacit [12] or difficult to articulate [13].-It can be individual and collective [14].-It can reside in people, groups, tools, tasks, and networks [15].

To recognize how innovation is produced in nature, we should explore first how living systems learn to adapt in their environments. Biologists Maturana and Varela introduced the concept of “autopoiesis” as the capacity of a living system to produce and maintain itself by creating its own parts, while constantly interacting and exchanging information with its ecosystem (structural coupling) [16].

This co-evolutional process lies at the basis of co-adaptation, intended as a form of cognition because the system “learns” how to be more suitable and respond better to its ecosystem, while the ecosystem learns how to adapt itself according to the changes of the system (Figure 1).

Biology helps to recognize that knowledge is always a co-creative process co-developed through “open” relationships, and, because the requirement of survival for living systems is the constant co-adaptation to the stimuli of internal and external environment, cognition lies at the basis of life itself.

In a similar fashion, the effectiveness of innovation processes depends in large part on the ability to combine market orientation (what is asked for) with effective organizational adaptability (matchmaking) to facilitate the emergence of different cognitive forms (market-based, scientific, technical, organizational, etc.).

According to the Resource Based Theory [18], relationships are powerful corporate resources; therefore, an organization aware of its core competencies can increase its creative potential by relating with others to generate new knowledge (Cognitive Processes Generativity) [19].

Knowledge is the main lever of an organization’s competitive advantage. It also includes the company’s off-the-shelf services (e.g., a clinician’s tacit skills are difficult to replicate externally). Consequently, biomedical organizations can valorize existing knowledge as well as generate new knowledge of strategic alliances.

### 2.2. Why Opening Is Fundamental for Research

In 1997, Christensen demonstrated how successful and excellent organizations can do everything “right” and yet lose their market leadership—or even fail—due to the rise of new and unexpected competitors taking over the market. Then, he introduced the concept of disruptive innovation by indicating a process of discontinuity that breaks with the status quo, accelerating the need for a creative response to the context [20].

However, the rigid and structured in-house innovation process assumes that successful innovation requires control and exclusivity. Technological, and scientific progress are evolving constantly in the biomedical field; therefore, a closed approach to innovation is no longer sustainable in an increasingly competitive, globalized, and dynamic scenario.

Producing scientific results and health services implies generating meaning, because such results help make sense of existence and life in the dialogue between different stakeholders [21]. Opening to new realities and languages becomes essential for the advancement of biomedical knowledge, and it is the pivotal theme of OI, a paradigm defined as “the purposeful use of internal and external knowledge to accelerate internal innovation and expand markets for the external use of innovation” [22].

This paradigm contrasts itself with the one described as closed, because it assumes that the locus of innovation may not necessarily be located within the organization but may also turn outward to a collaborative network that shares ideas and products through dynamic flows of change [23].

This innovative process is a result of the synergistic collaboration between the internal and external environment of the organization that valorizes its tangible (e.g., infrastructure, space, tools) and intangible (e.g., know-how, experience) assets. OI organizations leverage partnerships to access multiple resources; alliances can be combined in two ways to increase their potential value, as follows:-Knowledge integration: by internalizing the external resources they access with their efficiency and width (the more specialized they are, the less flexible they are).-Knowledge transfer: by constantly innovating their internal resources through specific tools that enable cross-fertilization of knowledge between organizations.

As a result, Gassmann and Enkel [24] identified three archetypes of OI:-Outbound: new ideas generated in the organization are disseminated outside, to other organizations and other environments (from inside to outside).-Inbound: knowledge within the organization is enriched with the integration of external resources (from outside to inside).-Combined: a balance is created between the two previous archetypes, in alliance with partners.

Much emphasis is given to individuals by recognizing their expertise, experience, and excellence. At the same time, organizations are encouraged to join international research and industry projects to forge meaningful collaborations, publish results, disseminate best practices, and transfer knowledge, tools, and methodologies.

### 2.3. From Theory to Practice

As stated before, knowledge innovation is a metamorphic process that arises from a difference in potential between what the ecosystem “requires” and the inner state of the organization (resources, skills, etc.). Innovation can be dictated by a sudden urgency (such as that experienced by hospitals in response to the recent pandemic, [25]), by challenges already present (the need to find cures for already existing diseases), or on the horizon (the possibility to develop novel treatments thanks to scientific/technological advancements).

Currently, several industries in the biomedical field are adopting the OI paradigm as a co-evolutional strategy. The integration of this paradigm within the field occurs between two unprecedented historical events. The first one was digital transformation, which involved a progressive integration of services and systems with novel digital solutions. Because the incorporation required professionals acquiring new skills and literacy—even for PM experts—a closed approach seemed to be less functional to compete in a complex and fast-changing world.

The second event that accelerated the need for outward openness was the COVID-19 pandemic, because it forced organizations to adopt innovative strategies (including boosting digital transformation) to survive and thrive. This event caused disruptive changes that pushed the world to react rapidly to challenges, such as real-time decision-making and business continuity, in new collaborative ways that went beyond the organizational boundaries [26,27].

Indeed, knowledge transfer across organizations was an OI endeavor, defined as “the use of purposive inflows and outflows of knowledge to accelerate internal innovation, and expand the markets for external use of innovation, respectively” ([26]; [28], p. 1). Because the new healthcare scenario shows multilateral interdependence between organizations and their ecosystems, its challenges might be addressed with this interdisciplinary logic.

We hereby introduce our experience as a high-volume research hospital in Italy (one of the first countries hit by the pandemic), in managing transformation from PM to PDM, adopting an OI-based strategy and pathway.

## 3. Methodology

The Fondazione Policlinico Universitario A. Gemelli IRCCS (FPG) is a large-volume polyclinic in Rome (Italy) recognized in 2018 as research hospital for PM and innovative biotechnologies by the Italian Ministry of Health. Being a research hospital means that research and clinical care must have mutual implications, providing citizens with the best therapeutic solutions. 

For a broad vision of the context, the hospital counts seven clinical departments with over 270 units, 1581 total beds and 52 surgical theaters (two hybrid rooms), one digital pathology, one MRI-integrated radiotherapy and one digital PET-CT scan. In 2021, 94,669 patients were treated (of which, 43,722 with oncological diseases) while 68,533 surgical procedures and 270 transplantations were performed. Despite the pandemic, 3606 COVID-19 positive patients were discharged, and 188,682 swabs were made [29].

The variety of resources and knowledge available today calls for organized (technical and personal) knowledge management of the community involved in the development of innovative projects and services [30], such as the hospital. In this sense, the Gemelli Science and Technology Park (G-STeP) played a key role in the management of heterogeneous and complex tangible and intangible assets of the hospital, focusing, directing, and best-directing innovation and creative processes, as well as product development. Born in 2020 as a core project of the Scientific Directorate, G-STeP has 21 research core facilities characterized by qualified personnel and traditional and state-of-the-art instrumentation to develop research projects that require specific data collection, storage, analysis, processing, and interpretation services (whether omics, clinical, digital, etc.) [31]. A virtual walkthrough can be taken at: Gemelli Science and Technology Park [G-STeP] (https://gstep.policlinicogemelli.it – accessed on 15 June 2022).

Within G-STeP, the Gemelli Generator (https://gemelligenerator.it/it/ – accessed on 15 June 2022) research center, developed by a multidisciplinary team, was created to concretely support the clinical and scientific enhancement of the expertise, processes and the huge amount of data collected over the years in the data warehouse (which is similar to a database) through advanced data extraction, collection, and analysis services. Its mission is to support the different assets of healthcare by adopting a strategic model that involves multiple different professionals, from basic research to clinical trials to real-world data (RWD), and that focuses specifically on the outcomes of care achieved with each patient.

In this center, the development of algorithms capable of guiding clinical decision-making falls under the constitution of true advanced medical devices and therefore requires stringent validation and verification of its requirements to meet the standards stated in the current medical device regulation (MDR) [32,33].

In line with the abovementioned historical events, in July 2020 the Scientific Directorate has strategically integrated OI by establishing a dedicated unit with the intent of boosting (supporting at a higher, more strategic level) research activities in PM. The unit developed and implemented a systematic activity (the so-called innovation radar) to identify potential innovative ideas through a six-step method:Scouting of an internal or external idea to valorize.Identification of the potential industrial/academic partner/s (matchmaking).Creation of a protected environment for the brokerage of intellectual property (non-disclosure agreements).Definition of the co-creation and co-development projects (cross-fertilization).Implementation of the co-creation and co-development project.Creation of a go-to-market strategy shared with the partner(s).

In addition, to implement the OI logic within an interdisciplinary hospital, two levels of complexity need to be addressed first.

### 3.1. Integrating a New Paradigm within the Hospital Traditional Philosophy

A primary level of complexity arose with the introduction of a paradigm shift in the philosophy of the hospital for a sustainability-oriented innovation. In fact, every form of innovation in biomedical organizations can be understood as a “4P innovation” [33] because it promotes deep transformations in terms of position, paradigm, process, and product [34].

Consequently, organizational innovation was visible:-in theories and practices supporting individual and collective mental models (paradigm);-in the biomedical field because new forms of knowledge are transferred and integrated in it with original solutions and ideas (position);-in the ways, methodologies, and pathways towards which every form of knowledge is generated (process); and-in the outcomes of personalized research, prevention, and treatment that, in this sense, differ from traditional medicine (product).

### 3.2. Integrating New Resources, Skills, and Ideas without Losing Organizational Identity

Taking care of a research idea from this perspective means protecting it from its inception to its longitudinal outcomes. In fact, this framework adheres to the principles of the organization at all stages of collaboration with external partners (in terms of co-creation and co-development), since the early and pre-competitive phases. 

However, “open” does not mean “free”. As Bogers [35] noted, in any R&D collaboration imprinted on OI there is a relational paradox to consider, which is about sharing knowledge on the one hand and protecting it on the other. The management of relational balances could be another very complex aspect, especially when different actors are involved in the innovation chain. Therefore, it is mandatory to identify the perimeter of the innovation process, as follows.

-Clarify the ethical principles of the organization and its value network.-Define business priorities (value chain, end users, and stakeholders).-Maintain the identity of the hospital while allowing for soft integrations.

The innovation process must be protected by confidentiality agreements, and it is often coordinated by professional figures that ensure the brokerage and protection of intellectual property. To this aim, organizations train their personnel to monitor these interactions, sometimes with dedicated units or departments [36].

Innovating within organizations inevitably touches some dimensions that may be relevant to those who are part of it. This means modifying achieved balances, sometimes eliciting ambivalent emotional responses [21]. Unthought-of interactions can generate knowledge never seen before. Moreover, building a knowledge-based trust value network can reduce the risk of opportunistic behaviors.

Because global progress makes the biomedical market more competitive, the OI paradigm enhances the expression of the “organizational polymorphism” without losing the original identity, while letting the natural co-evolution process originate new forms of activities and, potentially, new businesses and markets in a sustainable manner, which is similar to the sympatric speciation (Figure 2).

## 4. Results

Since opening to other realities generates several changes, we identified three levels of transformation within our setup:-The researcher metamorphosis (micro level); change happened within the core value of the organization; people and paradigm.-The organizational metamorphosis (meso level): individual change was translated into organizational practices and processes, leading the hospital to a new competitive position in the market.-The market metamorphosis (macro level): new digital and technological solutions have been implemented for PM to anticipate disease manifestation/progression/recurrence (e.g., predictive models).

### 4.1. The Researcher Metamorphosis

As stated previously, digital transformation required professionals to learn new skills and wording for understanding multidimensional data-driven predictive models dedicated to the personalization of clinical outcomes, leading to multiple challenges for adult learning [38]. In fact, many of them have been traditionally educated with strong specialist curricula; hence, it is more difficult to learn new abilities and transform their expertise without losing their professional identity. For this reason, OI has a strong pedagogical vocation because it recognizes the learning process being a transformation that involves aptitudes, skills, and competencies [39,40].

In order to facilitate the individual metamorphosis, it was essential to assess the following.

-What skills does the researcher already have?-What are the new skills the researcher needs to acquire?

This preliminary assessment was fundamental:-to recognize implicit and explicit know-how;-to create literacy (also when the researcher is highly educated and skilled); and-to train the researcher’s soft skills.

Indeed, the researcher metamorphosis is not just an individual but also an organizational phenomenon (context-based learning) [41]. For this reason, the Open Innovation Unit facilitated the change by estimating what skills the researcher needed to develop in a four-step process:Modeling: The unit performed the action to be learned (e.g., a first contact interview with an industry) while the researcher observed it.Coaching: The researcher is introduced to the subject/skill and assisted by the unit that provided feedback on what has been done.Scaffolding: The researcher learned to perform the task under the guidance of the unit.Fading: The unit continued to accompany the researcher allowing her/him to act independently and only providing support, if needed.

When mentorship was necessary, customized coaching sessions helped the researcher developing ad hoc entrepreneurial skills (e.g., networking, maintaining a growth mindset, communication, work ethics, etc.) [42], and acquiring a new level of professional identity (Figure 3).

As a result, from over 50 working groups with our researchers and their teams, we signed 35 non-disclosure agreements with several industries (pharma, biotech, venture capitalists, insurances, etc.) and signed four contracts for the co-creation and co-development of industrial projects (Table 1).

### 4.2. The Organizational Metamorphosis

Before embracing any project under the OI framework, the ecology of organizational transition should be carefully matched with, at least, two parameters:-Preparedness (paradigm-position): what are the dynamic capabilities* of the organization? Is it ready to deal with change and openness?-Readiness (process-product): what infrastructure (tangible** and intangible*** assets) does the organization have? Is it ready to support the innovation process?

After an in-depth assessment, the organizational metamorphosis was accompanied by harmonizing three parallel steps:a.The creation of a value network that respects the organizational identity, including ethical and scientific impact.b.The valorization of internal tangible** (e.g., facilities, instruments, technologies) and intangible*** (e.g., know-how, experience, expertise) assets. For example, we gave value to the expertise of our researchers in the conduction of clinical trials, as well as to the presence of the G-STeP with, in the case of digital medicine, the Gemelli Generator research center, specifically dedicated to data sciences.c.The habilitation of its dynamic capabilities, which consist in the ability to absorb, adapt, and build internal and external resources to rapidly respond to the changing environment [43].

By starting formal collaborations and cross-sectoral alliances to achieve deeper knowledge in PM, the strategic liaisons provided by the OI framework managed by the OI Unit of the Scientific Directore @FPG led to the results reported in Table 2 within two years.

Interdisciplinarity played a crucial role for learning adequate responses that encountered the unmet needs of the biomedical field as well as the necessity of the hospital to evolve with a sustainable effort.

### 4.3. The Market Metamorphosis

The boost of digital medicine has accelerated the global research process toward nonlinear dynamics through which various actors (researchers, clinicians, engineers, patients, etc.) interact and generate new knowledge and solutions.

To synergize this scientific development with the market progress, in October 2021 the FPG created a dedicated vehicle called Gemelli Digital Medicine & Health (GDMH) that fully exploits the hospital’s capacity and capabilities in the field of DM and health. Because FPG has a strong pedigree in PM research and care, GDMH helps generate, implement, and deploy DM and health services and solutions for PM worldwide with full ethical, technical, and regulatory credentials, fully compliant by design with the General Data Protection Regulation (GDPR) EU n. 2016/679.

From the perspective of OI, the organizational polymorphism of FPG has been respected by generating a new vehicle (GDMH)—conceived of as a sympatric speciation—belonging to the hospital and still powered by an OI logic, to scale-up the success story of FPG within the field of Digital Medicine and Health.

## 5. Discussion

Far from a reductionist approach, we can imagine the traditional ecosystem of PM drenched with heterogeneous data, whose analysis was following two main avenues:The study of -omic information (e.g., genomics, proteomics, metabolomics, etc.) analyzed by the discipline of bioinformatics to retrieve -omic patterns (for example, to identify a gene alteration and consequently find a personalized target therapy for that alteration) [44].The study of the clinical phenotype combined with holistic information (e.g., lifestyle, psychology, sociology, which go under the definition of RWD) analyzed by systems medicine and accelerated with the discipline of artificial intelligence to find a clinical pattern associated to a behavioral one, to predict and model their evolution into a clinical outcome and/or behavioral change [45].

Indeed, the integration of systems medicine with technologies and AI draw the line for understanding noncommunicable chronic diseases [46,47,48], for personalizing clinical phenotype when symptoms seem to be the same in multiple pathologies or when disease pathways can be several [1] and the management of multimorbidity [49]. 

DM brings these avenues together and their convergence generates the emergence of new actionable knowledge, meaning that both types of data (from -omics sciences and RWD) and their analysis is fundamental to produce a digital specimen (a digital biomarker, a predictive model or, on a higher hierarchy, a human digital twin) to be used in patient support programs up to the deployment of intelligent bot-based virtual coaching systems and the development of digital therapeutics, in silico clinical trials, etc. In this scenario, PM becomes PDM. The transition from one to the other occurs when researchers and organizations are ready to enhance their skills by welcoming new competences and understanding the implications of the digital transition in their practices and epistemologies.

Along these lines, we may indeed acknowledge that some challenges still occur, such as:-Strengthening the ethical and regulatory framework on data protection, privacy compliance and consent to treatment [50].-Expanding collaboration among different research centers and healthcare providers to share data [51].-Certifying/validating the reliability of data sources [52].-Implement a transparent and secure supply chain for collection, simplification, and conversion of raw to refined data for their actionability in clinical decision support systems that are useful for patients and physicians and, at large, for healthcare operators [53].-Implement an evidence-based supply chain on training, verification, and validation of algorithms in compliance with national and international laws and regulations [54].

## 6. Beyond Hype

Biomedical organizations and researchers interested in PDM that are willing to embrace the OI paradigm should consider that the scalability of an OI strategy depends on the structure and dynamics of the organization as well as the relational/territorial network in which it is located.

Socioeconomic determinants are also important in terms of innovation priority, to provide and give equal access to PM solution for patients and institutions; the cost of specific trials, target therapies, diagnostic tests, or information technologies could amplify the divide between those who can afford PM treatments and those who cannot [55]. PDM solutions (see Table 1) could help reduce costs for patients (e.g., travel expenditures), clinicians (e.g., time management), researchers (e.g., information sharing), and biomedical organizations (e.g., workflows optimization). However, the development and implementation of such approaches for disease prevention in public health is still in its infancy, although prevention plays a crucial role in health systems sustainability.

After assessing the potential outcomes of a P4 innovation (as described in Section 3.1) it is worth considering fostering a P4 education to generate authentic awareness among multiple actors such as patients, professionals, policy makers, and the public [56].

Relationships are also the basis of human development and care; therefore, they are essential for surviving but also for taking care of patients’ frailty. To this aim, PDM can really be conceived of as participatory if the personalization of DM—or better the digitalization of PM—could improve the quality of people’s lives as well as clinical practices, and if organizations and professionals are open to the dialogue and cross-fertilization with other disciplines, skills, and wisdoms. This enhanced personalized medicine would certainly go beyond the technological hype and base itself on the solid ground of bringing efficiency and accountability to the health systems.

To this aim, our article is an endeavor in confirming that using OI strategies could help managing environmental needs while achieving sustainability-oriented accountable innovation within biomedical organizations.

## Figures and Tables

**Figure 1 jpm-12-01500-f001:**
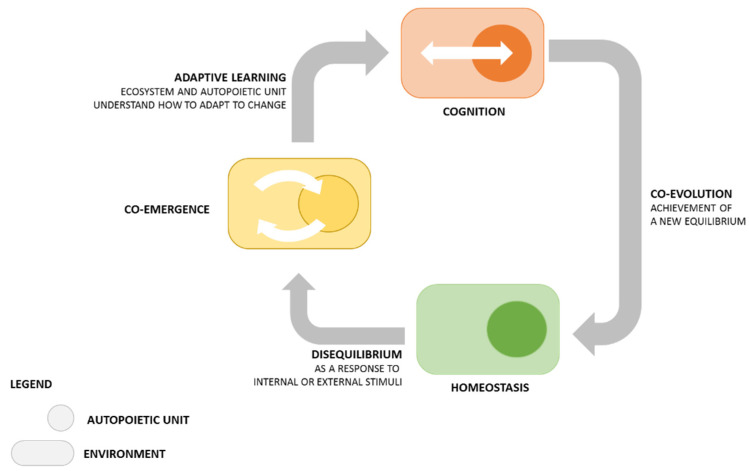
An autopoietic unit (living system) interacts with its environment (ecosystem) to maintain an equilibrium (homeostasis); hence, they organize each other by giving and interpreting reciprocal feedbacks (chemical, cognitive, etc.). While interacting (conceived of as a flow of stimuli), systems and ecosystems maintain their endogenous autonomy that differentiate them hierarchically and functionally. When a loss of balance is due to internal/external stimuli, the result of their interaction is a co-emergence of information because both parties need to re-establish their equilibrium. In order to achieve a new homeostasis, system and ecosystem must co-adapt to change intended as a learning experience. Once they “understand” how to reach a new balance (cognition), they co-evolve and therefore achieve a novel homeostasis. Because life itself is a continuous process between disequilibrium and homeostasis, Maturana and Varela draw the conclusion that “living is knowing and knowing is living” [16]. See also [17].

**Figure 2 jpm-12-01500-f002:**
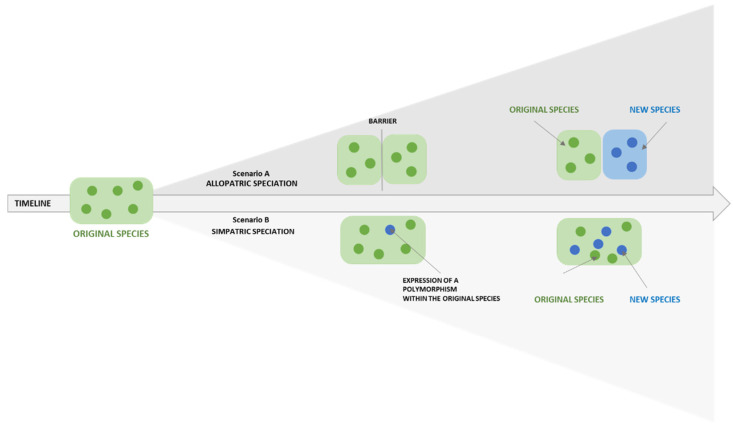
Speciation is an evolution process that generates a new species different from the pre-existent one. In synthesis, it can be allopatric (see Scenario A, a new species is created because of geographical barriers, meaning that a new business is generated but it is different from the original one), or sympatric (see Scenario B, a new species is created because of variation polymorphism within the pre-existent one, meaning that a new business is generated in harmonic integration with the original one). See also [37].

**Figure 3 jpm-12-01500-f003:**
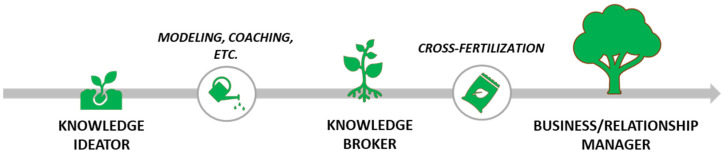
The metamorphosis of professional identity grows from being a thinker that generates knowledge and ideas to a progressive broker and manager of knowledge and business relationships.

**Table 1 jpm-12-01500-t001:** Example of digital medicine products.

Product	Short Description
Patient Support Program	Virtual programs through which care continuity can be provided to patients beyond the outpatient setting. By making use of multiple remote monitoring tools (apps, wearable devices, questionnaires, etc.) provided to the patient, healthcare professionals can check his/her health status in real time, be in constant contact with him/her, and intervene when appropriate [5]. AI algorithms can predict the risk of relapse (either physical or psychological), and they are specifically designed to monitor patients in the time distances between two clinical encounters (i.e., when he/she is not physically present in the hospital and may live outside the region). Patients are trained through virtual coaching sessions with dedicated health professionals (e.g., psychologist, nutritionist) in properly using the monitoring tools, and have access to apps/portals for educational materials on disease management. Those plug-ins are digital solutions aiming at meeting several needs in the patient journey and delivering innovative care models and/or identifying profiles of progression in the transition from health to disease.
Virtual Ward	Digitalized solutions for patient monitoring to facilitate real-time observation of certain parameters (clinical, psychological, etc.), to follow patients who cannot go to the hospital remotely [6]. For example, if a cancer patient is unable to travel from home (for example, due to an infection such as COVID-19), he/she can still be followed by doctors by checking his/her own biometric parameters (e.g., oxygen levels) after being adequately informed and trained for the correct use of the measuring tools. Information is delivered (via video call, call, or messaging) to health professionals who are in daily contact with the patient, offering telemedicine services and recalling him/her to the hospital for observation or treatment if necessary. Algorithms can support data processing to predict the future course of the patient’s condition and help the physician to make more personalized data-driven decisions.
Human Digital Twin	Computerized avatars that simulate the information of the patient, by connecting medical history (including family history, if available) with current illnesses and symptoms [7]. Human digital twins (HDTs) are created by mathematical and computational models to provide insightful mechanisms by regulating patient responses to treatment, and to generate virtual cohorts with practical applications in oncology (e.g., breast cancer, melanoma, brain cancer, lung cancer), infectious diseases (HIV, SARS-CoV2), metabolic diseases (diabetes) and cardiovascular diseases (e.g., coronary artery disease). HDTs are important for clinicians and patients because they give a comprehensive overview of the patient’s past and current clinical history, to develop a personalized plan and simulate predictions to anticipate disease onset or exacerbation.
In silico Clinical Trial	Reliable computational models of virtual trials, which can be used to study the effect of virtual treatments on virtual patients and predict the outcome of a clinical treatment in terms of safety and efficacy [8]. They can identify worst-case scenarios to optimize the safety of a preclinical trial. For example, machine learning (ML)/deep learning (DL) algorithms can be used in designing and simulating virtual trials—often customized on HDTs—to study the development or regulatory evaluation of a virtual drug, a device, or a therapeutic intervention. Studies with real patients may be reduced in favor of sophisticated simulations that predict, for example, the safety and efficacy of a treatment on a specific patient, or a subset of patients with similar clinical patophenotype. Another example regards the development/improvement of drugs and therapies to predict their outcomes (including the assessment of possible toxicity or the onset of adverse events) and personalize them according to each patient’s characteristics. These models can further facilitate greater understanding of molecules’ behavior (e.g., potential antimicrobial activity) by screening a large volume of molecules and virtually test them to identify antibacterial compounds structurally distant from known antibiotics.
Decentralized Clinical Trial	Studies that leverage “virtual” tools, such as sensory-based technologies, wearable medical devices, home visits, patient-driven virtual health care interfaces, and direct delivery of study drugs and materials to patients’ homes. In a fully decentralized clinical trial, subject recruitment, delivery and administration of study medication, and acquisition of trial outcomes data all proceed without involving in-person contact between the study team and the patient/subject [9]. Focused on patient-centricity, these trials allow patients to participate and improve compliance where constraints, such as socioeconomic factors, may occur by giving them more flexibility in ways to participate.
Digital Therapeutics	Evidence-based therapeutic interventions for the patient that a physician “prescribes” from a clinical dashboard (e.g., alerts for therapy management, progress tracking, health literacy, doctor jump-in) based on AI algorithms that are driven by high-quality software programs to prevent, manage, or treat a medical disorder or disease. They are used independently or in concert with medications, devices, or other therapies to optimize patient care and health outcomes [10]. The “active ingredient” corresponds to the component that shows an evidence-based therapeutic effect (e.g., motivational interview, psychoeducation, or similar tools elaborated on the experience of clinicians or from the literature). The “excipient” shapes the active ingredient to promote its intake; therefore it can be the user interface, specifically designed to maximize the efficacy of the active ingredient.

**Table 2 jpm-12-01500-t002:** Results from the OI interdisciplinary activities with G-STeP/Gemelli Generator and other units at FPG (July 2020–July 2022).

Results	Details	Total
Fundings	Fundings for industrial projects Contracts for industrial projects (FPG)	497,000 € 4
	Fundings from European projects European projects awarded	1,723,250 € 4
	Funding for Innovation Total funds raised	12,000,000 € 14,220,250 €
Projects	Patient support programs - Of which virtual wards	3 1
	AI-driven predictive models Investigator-driven clinical trials	1 4
Publications *	Peer-review articles	31
	Books	3
	Chapters Total impact factor Total citations	14 91.6 points 71
Education	Training courses provided by the OI Unit - Of which for universities - Of which for medical specializations - Of which for industries - Of which for clinical research	10 3 3 3 1
	Our OI Unit as a “case study” for Master thesis	2

* It is worth mentioning that a brand-new editorial line has been developed to raise awareness on the topic of PM among different stakeholders, as a co-emergence between the main focus of our research lines and the new OI approach introduced.

## Data Availability

Not applicable.

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
