# Peer review of "Open Innovation as the Catalyst in the Personalized Medicine to Personalized Digital Medicine Transition"

_jpm, 2022, doi:10.3390/jpm12091500_

Round 1

Reviewer 1 Report

The authors reported an interesting topic of transition from Personalized Medicine to Personalized Digital Medicine. Minor comments need to be addressed before it is accepted for publication.

-          It is recommended to add the abbreviations after the first time of using the term. For example, Table 1 HDT

-          Grammar check is recommended for example in Table 1 authors states “HDT are important for clinicians and patients because they give a comprehensive overview of the patient past and current to develop a personalized plan and simulate predictions to anticipate disease onset or exacerbation.” – patient past and current ….?…the sentence needs to be checked

Author Response

Dear Reviewer,

thank you for your kind suggestions that we addressed by completing the sentence in Table 1 as well as reporting the extended term for Human Digital Twins before the abbreviation.

The Authors

Reviewer 2 Report

In Table 1, the listed items are really services - should give examples of actual products for each service.

Personalized medicine v. public health tradeoffs can be discussed in terms of innovation priority - e.g., is cost an issue.

Include examples of 'personalized innovation'  -accelerated  pharma discovery/inventions, target drug therapy, e.g., in cancer.

Author Response

Dear Reviewer,

thank you for your kind suggestions, which we addressed as follows:

  • We integrated some examples for each solution in Table 1.
  • We provided an example of innovation in personalized medicine [lines 68-70] and included a dedicated reference.
  • We further discussed how innovation in Personalized Medicine can lead, for example, to a divide between people and organizations who can afford expensive target therapies and those who cannot [lines 442-451], and we added a new reference.

The Authors